# The Impact of Depth and Width on Transformer Language Model Generalization

## Abstract

Transformer language models tend to perform better the more parameters they have. Previous theoretical and empirical work suggests that the total number of parameters is not the only relevant factor, however; rather, expressivity and out-of-distribution generalization may benefit more from increasing depth than increasing width. To test this hypothesis we disentangle depth from the number of parameters, constructing families of models which trade off depth for width while keeping the total number of parameters constant. We pretrain those models and evaluate them on both language modeling and compositional generalization tasks. We report three main conclusions: (1) within each family, deeper models show better language modeling performance, but the relative benefit of additional layers diminish rapidly; (2) when fine-tuned on compositional generalization tasks, deeper models generalize better out-of-distribution than shallower models do, but returns are similarly diminishing; (3) the benefits of depth for generalization cannot be attributed solely to better performance on language modeling or in-distribution data. These results replicate in three different model families (41M, 134M and 374M parameters), suggesting that depth improves performance across model sizes.

## 1 Introduction

The number of possible sentences in natural language is enormous; regardless of the size of its training set, a language model (LM) will regularly encounter sentences it has never seen before. The ability to interpret such sentences relies on compositional generalization: the capacity to combine familiar words and syntactic structures in new ways (Montague, 1970; Fodor & Pylyshyn, 1988). Transformer LMs (Vaswani et al., 2017), while highly successful in many settings, often struggle when tested on benchmarks that require compositional generalization (Kim & Linzen, 2020). What architectural factors affect a transformer's ability to generalize compositionally?

In this paper, we test the hypothesis that increasing a transformer's depth—the number of layers it has—improves its out-of-distribution performance on tasks that require compositional generalization. This hypothesis is motivated both by theoretical work, which has shown that adding layers increases the expressive capacity of neural networks in general (Raghu et al., 2016) and transformers in particular (Merrill et al., 2021), and by experimental work suggesting that deeper models may generalize more compositionally than shallower ones (Mueller et al., 2022; Murty et al., 2022).

While existing work lends credibility to this hypothesis, directly confirming it requires addressing the confound between depth and size (number of parameters). As each additional layer introduces a new set of parameters, deeper models are also larger, all else being equal. LMs' performance on a wide variety of tasks is strongly correlated with the number of parameters they have (Kaplan et al., 2020; Hoffmann et al., 2022; Muennighoff et al., 2023). To disentangle these two factors, we construct classes of models with equal numbers of total parameters but differing depths; we do so by reducing the model's feed-forward dimension to compensate for added depth. We pretrain all models on a language modeling task, and fine-tune them on four compositional generalization tasks: COGS (Kim & Linzen, 2020), COGS-vf (Qiu et al., 2022a), GeoQuery (Zelle & Mooney, 1996), and the English passivization portion of Multilingual Transformations (Mueller et al., 2022).

In addition to any possible direct effect on compositional generalization, depth may also be correlated with other factors which may themselves predict out-of-distribution generalization, such as language modeling loss during pretraining or in-domain fine-tuning performance. This complicates

the interpretation of any relationship we might find between depth and generalization performance. To address this concern, we investigate and correct for the effect of depth on language modeling performance and in-distribution loss. We report the following findings, which hold across three model size classes (41M, 134M, and 374M parameters):

1. In general, deeper models have lower perplexity (Section 3.1). The marginal increase in performance gained by adding more layers diminishes rapidly as models get deeper, and performance begins to degrade when the feed-forward dimension approaches the dimensionality of the model's contextualized embeddings.

2. In general, deeper models generalize better on tasks that require compositional generalization (Section 3.2). Again, most of the benefit of depth accrues from the first few layers; for several of the tasks, performance saturates very quickly as models get deeper.

3. Deeper models generalize more compositionally even after correcting for the fact that their language modeling perplexity is lower and their in-distribution performance higher (Section 3.3).

## 2 METHODOLOGY

### 2.1 CONSTRUCTING FAMILIES OF MODELS WITH EQUAL NUMBERS OF PARAMETERS

To make a transformer LM deeper without increasing the total number of parameters, we need to also make it narrower. There are several ways to do so: we can reduce the size of the feed-forward dimension $d_{\text{ff}}$, reduce the size of the residual stream (the embedding size) $d_{\text{model}}$, or reduce the size of the attention outputs $d_{\text{attn}}$ (see Appendix B for a diagram of a transformer layer annotated with dimensionality labels). Vaswani et al. (2017) coupled these three variables at $d_{\text{model}} = d_{\text{attn}} = d_{\text{ff}}/4$. Most transformer LMs have adopted this ratio (Devlin et al., 2019; Kaplan et al., 2020; Hoffmann et al., 2022, *inter alia*), though Raffel et al. (2019) increased the size of $d_{\text{ff}}$ relative to $d_{\text{model}}$ and $d_{\text{attn}}$ for their two largest models. By contrast, we vary $d_{\text{ff}}$ with depth (while holding $d_{\text{model}} = d_{\text{attn}}$ constant). By keeping the attention mechanism identical across models of varying depths, we rule out the possibility that model depth will be confounded with the capacity of a model's self-attention mechanism. We refer to $d_{\text{model}}/d_{\text{ff}}$, conventionally set to $1/4$, as the *feed-forward ratio*.

**Deriving hyperparameter relations** As a starting point for our size classes of models, we use hyperparameters taken from the T5-base and T5-large size classes (Raffel et al., 2019) as well as a smaller model from Kim & Linzen (2020) which has identical layer-internal hyperparameters to T5-small but has fewer layers. We then calculate how much the size of the feed-forward dimension must change to accommodate adding or removing layers. Starting from the parameter formula in Kaplan et al. (2020), the number of parameters $M$ in a single layer is

$$M(d_{\text{ff}}) = 2d_{\text{model}}d_{\text{ff}} + 4d_{\text{model}}d_{\text{attn}} = \beta \cdot d_{\text{ff}} + A,$$

where the constant $\beta$ represents the contribution of the parameters of the feed-forward block which project vectors from $\mathbb{R}^{d_{\text{model}}}$ into $\mathbb{R}^{d_{\text{ff}}}$ and back into $\mathbb{R}^{d_{\text{model}}}$; and the constant $A$ represents the parameters of everything aside from the feed-forward block, including the attention mechanism.[1] The total parameter count of a full model $N$ in terms of $d_{\text{ff}}$ and $n_{\text{layers}}$ is then

$$N(n_{\text{layers}}, d_{\text{ff}}) = n_{\text{layers}} \cdot M(d_{\text{ff}}) + 2d_{\text{model}}n_{\text{vocab}} = n_{\text{layers}} \cdot M(d_{\text{ff}}) + E,$$

where $E$ represents the parameters of the vocabulary embedding and unembedding transformations. Given initial values $(n_{\text{layers}}^0, d_{\text{ff}}^0)$ which characterize the baseline model in each size class (e.g., T5-large), our goal is to find pairs $k, w(k)$ such that

$$N(n_{\text{layers}}^0 + k, d_{\text{ff}}^0 - w(k)) = N(n_{\text{layers}}^0, d_{\text{ff}}^0).$$

Solving for $w$ as a function of $k$ tells us how much to increase (or decrease) $d_{\text{ff}}^0$ if we remove (or add) $k$ layers from an existing model:

$$w(k) = \left\lfloor \left(1 - \frac{n_{\text{layers}}^0}{n_{\text{layers}}^0 + k}\right)\left(d_{\text{ff}}^0 + \frac{A}{\beta}\right) \right\rfloor, \tag{1}$$

---

[1]The number of attention heads does not affect the parameter count, only how the existing attention parameters are partitioned among each head.

| | 41M | | | | | | | 134M | | | | | | | 374M | | | | | | |
|---|---|---|---|---|---|---|---|---|---|---|---|---|---|---|---|---|---|---|---|---|---|
| $n_{\text{layers}}$ | 1 | **2** | 3 | 4 | 5 | 6 | 7 | 1 | 2 | 6 | 8 | **12** | 16 | 21 | 1 | 2 | 4 | 6 | 8 | 12 | 16 | **24** |
| $d_{\text{ff}}$ | 4779 | **2048** | 1138 | 682 | 409 | 227 | 97 | 36k | 17k | 5121 | 3584 | **2048** | 1280 | 731 | 99k | 49k | 24k | 15k | 11k | 6998 | 4907 | **2816** |
| | $d_{\text{model}} = d_{\text{attn}} = 512, \; n_{\text{heads}} = 8$ | | | | | | | $d_{\text{model}} = d_{\text{attn}} = 768, \; n_{\text{heads}} = 8$ | | | | | | | $d_{\text{model}} = d_{\text{attn}} = 1024, \; n_{\text{heads}} = 64$ | | | | | | |

Table 1: Models of varying depths across three size classes. Bolded variants are the baseline models whose hyperparameters were taken from Kim & Linzen (2020) and Raffel et al. (2019).

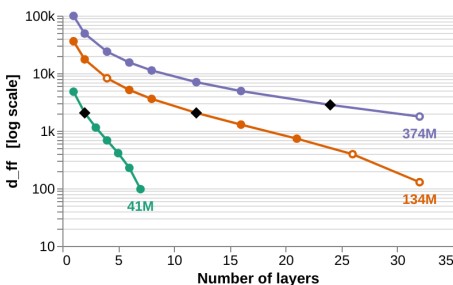

41M

134M

374M

Figure 1: Models for the 41M-, 134M-, and 374M-parameter size classes. Points indicate models trained in this paper, and black diamonds represent the baseline models for each class whose hyperparameters were taken from Kim & Linzen (2020) and Raffel et al. (2019). Unfilled points indicate models which were only pretrained, not fine-tuned.

Since adding or removing $k$ layers might require changing $d_{\text{ff}}^0$ by a fractional amount, we round $w(k)$ to the nearest integer; this means that are models may not be exactly equal in total parameter count, but the differences are very small relative to $N$. Table 1 reports the exact hyperparameter values we use for each of our three size classes, derived from Equation 1 above. Figure 1 shows each size class plotted as $(n_{\text{layers}}, d_{\text{ff}})$ pairs. We note that in some cases this manipulation results in models with a feed-forward ratio of greater than 1, that is, in models where $d_{\text{model}} > d_{\text{ff}}$; we explore the implications of such ratios in Section 3.1.

## 2.2 DATASETS AND TRAINING

### 2.2.1 LANGUAGE MODELING

We use the Colossal Clean Crawled Corpus (C4; Raffel et al. 2019) as our pretraining corpus. C4 was created by filtering data from the Common Crawl dataset of scraped web files. We pretrain each model on a causal language modeling objective using the C4 corpus. We use a context size $n_{\text{ctx}}$ of 1024 tokens and a batch size of 128 sequences $\approx$ 131k tokens. We pretrain each model for 1M steps, resulting in a total training dataset of roughly 131B tokens.

### 2.2.2 COMPOSITIONAL GENERALIZATION

In compositional generalization datasets, models are tested on a distribution which contains novel combinations of pieces which have been previously seen independently during training. We fine-tune our pretrained models on the training portion of the dataset for 10,000 steps, measuring in-distribution generalization accuracy (validation accuracy) every 250 steps. Validation loss continued to decrease throughout training runs on each dataset, so we report values from the end of each fine-tuning run without early stopping. We use four compositional generalization datasets (for examples of instances of these tasks, see Table 2):

1. **COGS** (Kim & Linzen, 2020) is a semantic parsing dataset introduced to serve as a test for compositional generalization. It consists of natural-language sentences paired with formal semantic representations, and is constructed such that the out-of-domain generalization distribution contains two generalization types: new combinations of familiar words (*lexical generalization*, such as using the word 'hedgehog' as the object of a sentence when it has only been seen during training as a subject); or using known words in new syntactic structures (*structural generalization*, such as relative clauses that are more deeply nested than seen in training).

2. **Variable-free COGS** (COGS-vf; Qiu et al. 2022a) is a simplified variant of COGS where the semantic representations are converted into a form which does not use numbered variables (see Table 2 for a comparison between COGS and COGS-vf). Removing variables from the representation has the benefit of lowering the associated computational cost of training by making

| COGS | $x$: A hedgehog ate the cake . |
| | $y$: *cake$(x_4)$; hedgehog$(x_1)$ AND eat.agent$(x_2, x_1)$ AND eat.theme$(x_2, x_4)$ |
| COGS-vf | $x$: A hedgehog ate the cake on the bed . |
| | $y$: eat(agent = hedgehog, theme = *cake(nmod.on = *bed)) |
| GeoQuery | $x$: which states have cities named m0 |
| | $y$: answer(intersection(state, loc_1(intersection(city, m0)))) |
| English passivization | $x$: our vultures admired her walrus above some zebra . |
| | $y$: her walrus above some zebra was admired by our vultures . |

Table 2: Examples of inputs ($x$) & targets ($y$) from each compositional generalization dataset.

sequences meaningfully shorter. This conversion has been previously shown to improve the performance of models by reducing the complexity of the output space (Qiu et al., 2022b), but comes at the cost of limiting the capacity of the formal language to represent many phenomena in natural language which require coordination of variable identity, such as control and anaphor binding.

3. **GeoQuery** (Zelle & Mooney, 1996) contains natural-language questions about US geography paired with SQL-style database queries representing those questions. We report results on the GeoQuery Standard split.

4. **English passivization** (Mueller et al., 2022) is a dataset of English active-voice sentences paired with their passive-voice counterparts (adapted from Mulligan et al. 2021). This benchmark is designed to test whether models use shallow, positional heuristics or syntactically-sensible ones. While Mueller et al. (2022) implemented a number of transformations in different languages, we focus on the English Passivization task.

## 3 RESULTS

### 3.1 LANGUAGE MODELING

**Deeper models have lower perplexity.** Depth has a significant impact on model performance. At the shallow end of the spectrum, increasing model depth results in a dramatic improvement in perplexity (Figure 2). In Figure 3a we compare the perplexity of each model in a size class relative to that of the best-performing model of that size. In the extreme case, the perplexity of a single-layer model can be nearly twice that of the optimal model in the class. Moreover, as parameter count increases the disparity between the worse, shallower models and the better, deeper models increases as well: For 41M-parameter models the ratio between the perplexity of the single-layer model and that of the optimal (5-layer) model is 1.59; for the 134 M-parameter models, the ratio is 1.86; and for the 374M-parameter models, the ratio is 1.99.

**Performance increases most rapidly within the first few layers.** While deeper models do, in general, perform better than shallower ones, the increase in performance that comes from adding layers diminishes rapidly as models become deeper (Figure 3a). The performance difference between 1-layer and 2-layer models is dramatic across all size classes; moving from 2 to 4 layers results in a much more modest performance improvement. We also note that as models get larger in our setup, they are able to make productive use of increasingly more layers: the optimal 41M-parameter model in our setup has 5 layers, while the optimal 134M-parameter model has 12; among 374M-parameter models, the 24-layer model had the best performance. At the same time, the pattern of the diminishing utility of depth holds even for the largest models we study.

**Performance starts degrading when models become too narrow.** At the deeper end of our scale, adding layers is not only unhelpful for performance, but begins to harm it (see the right-hand sides of each size-class curve in Figure 3a). As previously established, the point at which trading width for depth becomes harmful is not an absolute function of depth, since the optimal models from each size class have differing depths. However, comparing the relative performance of models within a size class to the feed-forward ratio $d_{\text{model}}/d_{\text{ff}}$ shows that model performance begins to worsen once $d_{\text{ff}}$ becomes smaller than $d_{\text{model}}$ (to the right of the red dashed line in Figure 3b); when this happens, the

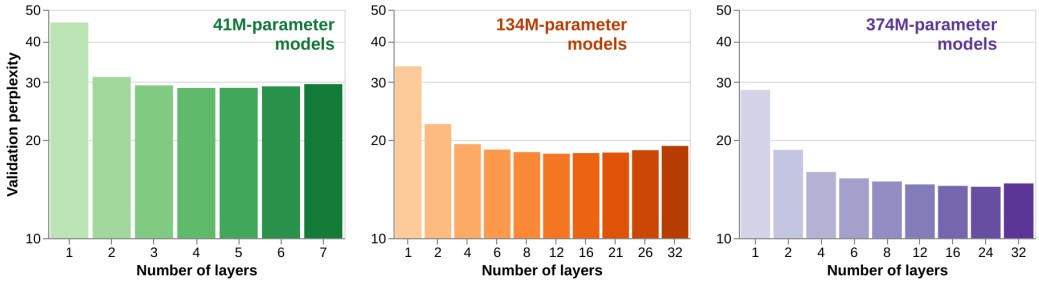

Figure 2: Deeper models achieve lower perplexities than shallower ones after equal amounts of training data regardless of size, but the benefits of adding layers diminish quickly with depth.

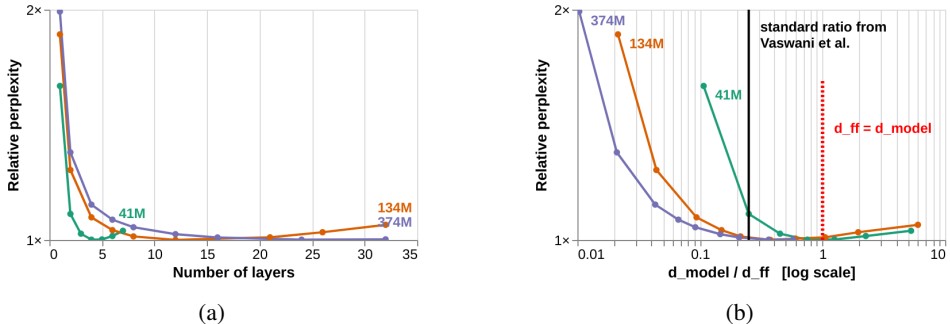

(a)                                                        (b)

Figure 3: Relative perplexity compared to the best model in each size class. **(left)** Perplexity goes down rapidly as models get deeper; only a few layers are needed to obtain most of the value of depth. **(right)** When $d_{\text{model}}/d_{\text{ff}} > 1$ (red dashed rule), perplexity slowly increases. As models get larger, the range of $d_{\text{model}}/d_{\text{ff}}$ ratios where performance is close-to-optimal expands leftward to include smaller and smaller values.

affine projection of the vectors from $\mathbb{R}^{d_{\text{model}}}$ into $\mathbb{R}^{d_{\text{ff}}}$ becomes a non-injective map. In Appendix C we analyze the weight matrices of the affine transforms in the feed-forward network of each layer and demonstrate that as $d_{\text{model}}/d_{\text{ff}}$ increases the transforms become increasingly rank-deficient.

**Larger models are more robust to changes in the feed-forward ratio.** Varying $d_{\text{ff}}$ while keeping $d_{\text{model}}$ constant results in feed-forward ratios $d_{\text{model}}/d_{\text{ff}}$ which deviate significantly from the standard ratio of $1/4$ (black vertical rule in Figure 3b). We find that smaller models are more sensitive to the particular value of the feed-forward ratio, and that for small models the standard ratio may not be optimal. Within the 41M-parameter size class there is a narrow range of feed-forward ratios in which model performance is within a few percentage points of the best-in-class model. As models get larger, this range expands leftward to include models which have increasingly wide feed-forward networks relative to the size of their contextual embeddings. This shows that larger models have more leeway to trade depth for width, becoming wider in proportion to their model dimension $d_{\text{model}}$ without incurring large penalties for their perplexity. It also shows that when $d_{\text{model}}/d_{\text{ff}} < 1$ the feed-forward ratio no longer serves as a predictor of relative perplexity independent of size.

## 3.2 COMPOSITIONAL GENERALIZATION

To test the impact of depth on compositional generalization, we fine-tune the models pretrained in the previous section on the training portions of each of the compositional generalization benchmark datasets. We measure the full-sequence (exact match) accuracy of the models on the out-of-distribution generalization set and note several findings:

**Deeper models generalize better.** As with language-modeling performance, depth has a significant impact on how well models generalize (Figure 4). On each of the datasets, deeper models tend to attain higher generalization accuracies than shallower models in the same size class. The

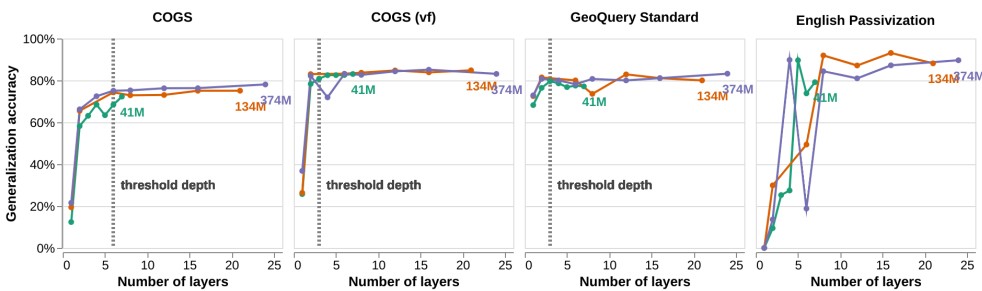

Figure 4: Deeper models generalize better than shallower models across datasets and size classes. Beyond the observed threshold depths on COGS, COGS-vf, and GeoQuery Standard, increasing depth does not affect model performance; these thresholds do not change as model size grows.

effect of depth on compositional generalization is more variable than it is for language modeling: for COGS, COGS-vf, and GeoQuery we note some small non-monotonicity in the generalization accuracy across different depths. On English Passivization, the 41M- and 134M-parameter model classes show largely-consistent trends where deeper models perform better than shallower ones; the 374M-parameter models do show more significant non-monotonicity, though the deepest models do still outperform the shallowest ones.

**The benefit of depth saturates quickly for some tasks.** As with language modeling, most of the benefit of depth is gained by having only a few layers. For three of the tasks—COGS, COGS-vf, and GeoQuery—we see threshold depths after which generalization accuracy stays relatively constant as depth increases. These threshold depths are low and constant across model sizes, but vary by dataset: 4–6 layers for COGS, and 2–4 layers for COGS-vf and GeoQuery. Performance on COGS-vf appears to saturate with fewer layers than on COGS despite the two datasets being equivalent in expressive capacity;[2] this suggests that the saturation we observe on some datasets is closely linked to the complexity of the output representation independent from the complexity of the compositional generalization expressed in the data. On English Passivization, the impact of depth is more variable; deeper models to generalize better than shallower ones, though we observe more noise between depths, which makes it difficult to ascertain if a size-independent threshold exists.

The threshold effects suggest that some subsets of the datasets can be addressed with relatively simple models. We investigate this hypothesis using the fact that COGS and COGS-vf include two types of generalization cases: lexical generalization, where a familiar word needs to be interpreted in a familiar syntactic context in which it has not been observed; and structural generalization, where the syntactic structure is novel and needs to be constructed from familiar syntactic pieces. Breaking performance down by the type of generalization required, we find that even deep models at the largest model size systematically fail to generalize structurally (Figure 5); the benefit of depth is largely limited to the easier lexical generalization. This supports the hypothesis that the saturated effect of depth is due to the existence of easier subsets of the datasets, and shows that increasing depth alone does not allow a model to learn the correct inductive bias for these structural tasks.

### 3.3 THE EFFECT OF DEPTH ON GENERALIZATION IS NOT SOLELY ATTRIBUTABLE TO BETTER PRETRAINING LOSS OR IN-DISTRIBUTION PERFORMANCE

Although deeper models generalize better than shallower models do, our pretraining analysis in Section 3.1 shows that deeper models also attain lower validation perplexities on their pretraining corpus than shallower models. Additionally, we observe that deeper models achieve lower in-distribution loss on the compositional generalization tasks than shallower models. These two results are potential confounds for the interpretation of the previous section: perhaps depth does not *directly* improve generalization accuracy, but only does so indirectly by allowing models to either become better LMs or else to better learn the in-distribution fine-tuning data. To determine whether that this is the case,

---

[2]As previously noted, COGS can represent phenomena that COGS-vf cannot, but both output representations are sufficiently rich to represent the examples studied here.

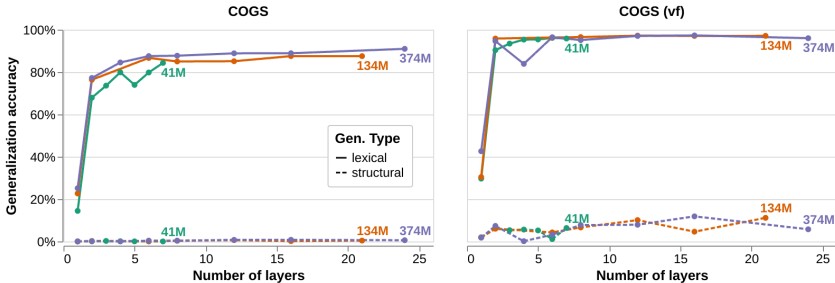

Figure 5: Increasing depth helps on the lexical generalization tasks (solid lines) in both COGS and COGS-vf, but does not meaningfully improve performance on the structural tasks (dashed lines).

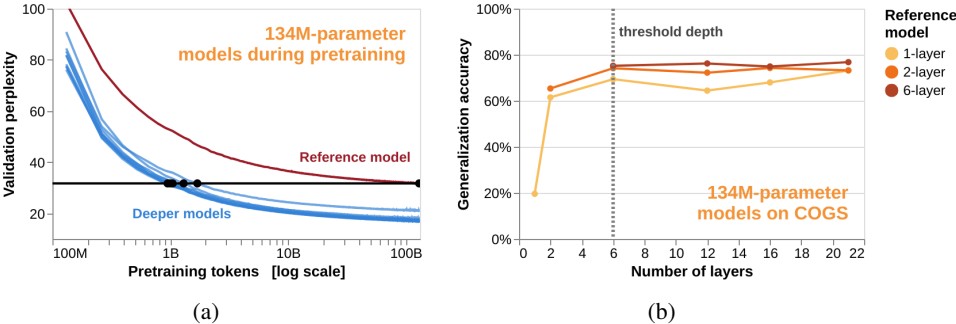

(a)                                    (b)

Figure 6: **(left)** To correct for the effect deeper models having better pretraining losses may have on generalization accuracy, we pick a reference model depth (red) and use checkpoints (black) from deeper models (blue) which have equal validation perplexity as the reference model does at the end of its pretraining. We then fine-tune these 'pretraining-corrected' checkpoints on the compositional tasks. **(right)** Even when fine-tuning checkpoints with equal validation perplexity, deeper models still generalize better than shallower models do up through six layers. The figure shows generalization accuracies from 134M-parameter models on COGS.

or whether depth does in fact directly improve generalization, we correct for both of these potential confounds.

First, to correct for the fact that deeper models attain lower pretraining losses, we repeated our fine-tuning experiments using checkpoints of models that have equal validation perplexities within a size class. We pick the least-performant (i.e., shallowest) model within a size class as the "reference model" and note its validation perplexity at the end of pretraining. We then pick the checkpoints of all deeper[3] models at the point when they achieved this reference perplexity (Figure 6a). Finally, we fine-tune each of these checkpoints on the compositional generalization tasks. We repeat this process for successively deeper reference models. We find that even when fine-tuning from checkpoints of equal validation perplexity, deeper models still generalize better than shallower models (Figure 6b). For compositional datasets where we observe thresholding behavior, the benefits of depth continue to hold up through that threshold depth.

Next, we correct for the potentially confounding fact that deeper models learn the in-distribution split of the compositional generalization tasks better than the shallower models do. To do this, we compare the generalization accuracies of models at points during fine-tuning when they have equal in-distribution loss. Figure 7 shows that even after adjusting for in-distribution performance, deeper models still achieve higher accuracies on the out-of-distribution generalization set than shallower models do.

---

[3]We only consider models deeper than the reference model since, in general, shallower models will never attain the perplexity of the reference model at the end of its pretraining. This assumption breaks down when considering the deepest models in each size class, but these are far deeper than the points at which depth seems to saturate performance on our compositional datasets so we do not extensively explore this regime.

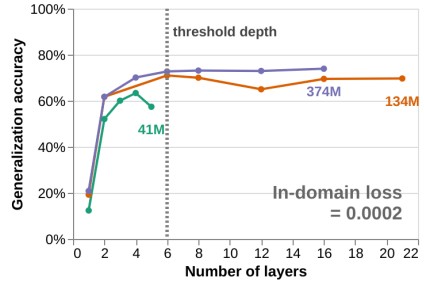

Figure 7: Deeper models generalize better than shallower ones on the COGS compositional generalization dataset, even at points during fine-tuning when models have equal loss (0.0002) on the in-distribution data.

## 4   RELATED WORK

**Compositionality**   Previous work has explored the degree to which neural models exhibit compositional behavior by training or fine-tuning models on compositional tasks such as simple command sequences (Lake & Baroni, 2018) or semantic parsing (Kim & Linzen, 2020; Keysers et al., 2020). Other work has explored methods to improve the compositional behavior of models, including through data augmentation (Qiu et al., 2022a), larger models (Qiu et al., 2022b), and architectural changes (Gordon et al., 2019; Csordás et al., 2021; Ontañón et al., 2021). Our work complements these approaches by exploring architecture changes which do not change total model size.

**Comparison to standard architectures**   We primarily focus on models that are shallower and wider than standard convention. Since $d_{model}$ is fixed within each class this means that most of our models have increasingly small feed-forward ratios $d_{model}/d_{ff}$; moreover, since $n_{layers}$, $d_{model}$, and $d_{ff}$ tend to increase in standard architectures as parameter count grows, this means that the disparities between our shallowest models and the conventional ones grows as the size class gets bigger. Exact parameter counts differ from the corresponding models in Raffel et al. (2019) and Kim & Linzen (2020) owing to differences in the size of the vocabulary/embedding layers and the fact that we use decoder-only models rather than encoder-decoder models, though the layer-internal hyperparameters of our base models are consistent with theirs. Qiu et al. (2022b) found that decoder-only models performed similarly to encoder-decoder models of comparable size; following Wang et al. (2022) we consider decoder-only models with half as many total layers as their encoder-decoder variants.

**Impacts of Depth**   Theoretical work has shown that the expressive capacity of neural networks in general (Raghu et al., 2016) and transformer models in particular (Merrill et al., 2021) grows exponentially in depth. Empirical work also points to the role of depth in model performance. In a more general setting, Tay et al. (2021) found that scaling by depth is generally more helpful than scaling by width on downstream tasks. For compositional generalization in particular, Mueller et al. (2022) found that reducing depth was more harmful than reducing with for pretrained encoder-decoder models. Murty et al. (2022) observed that deeper transformer encoders often have more tree-like representations and parsing accuracies on some compositional tasks. Tempering these positive results, Veit et al. (2016) noted that in models with residual connections, even very deep networks leveraged only shallow subnetworks of roughly constant depth. Brown et al. (2022) also concluded that wide, shallow transformer models can attain roughly-equal performance to deeper ones. Both sets of results, however, are confounded by a lack of control for total parameter count.

**Controlling for model size**   There are various choices to be made when studying the the impact of various hyperparameter choices without affecting the net model size, i.e constructing size classes of models. Kaplan et al. (2020) covaried the number of layers $n_{layers}$ with the contextual embedding dimension $d_{model}$, which they coupled to the attention-internal $d_{attn}$ and feed-forward dimention in the standard ratio of $d_{model} = d_{attn} = d_{ff}/4$. Among models of an equal size, they concluded that performance increases are largely driven by increasing the total parameter count of models, and that within "reasonable limits" language modeling perplexity is only weakly dependent on shape (though Tay et al. 2021 concluded that the same was not true for performance on downstream tasks, but did so without controlling for the impact of size).

## 5 LIMITATIONS & FUTURE WORK

**Number of runs**   Due to compute limitations, the results we report represent a single pretraining and fine-tuning run for each condition. Given the fact that out-of-distribution generalization in fine-tuning is particularly sensitive to random seeds (McCoy et al., 2020), multiple runs for each condition would decrease noise in the results (Figure 4), increase our confidence in the effect sizes we report, allow us to quantify our uncertainty over them, and extend our results on generalization to include the deepest, narrowest models.

**Alternative approaches to controlling for total size**   Our approach to controlling for total parameter count necessitates making depth-width trade-offs. An alternative approach would be to construct Universal Transformers (Dehghani et al., 2018), where each model in a size class has a transformer layer with the same parameters repeated $n_{\text{layers}}$ times. Such a weight-sharing approach would allow for deeper models to have arbitrarily-wide feed-forward networks, mitigating the impact of making models too narrow. While such weight sharing prevents models from performing different computation in different layers, such restriction may in fact be beneficial for compositional generalization where similar computations (e.g., combining two syntactic phrases to a larger phrase) may need to apply recursively at different scales.

**Pretraining corpus effects**   We consider models pretrained on natural-language data. For our particular choice of compositional generalization experiments, the presence of lexical items in both the pretraining corpus and the generalization datasets represents a potential confounder of generalization performance which could be mitigated by modifying compositional datasets (Kim et al., 2022). More generally, the distribution of pretraining data affects the inductive biases conferred to LMs (Papadimitriou & Jurafsky, 2023). As a particular area of interest for future work, we point out the hypothesis that including source code in the pretraining corpus (OpenAI, 2023; Google, 2023) will improve compositional generalization.

**Fine-tuning vs. in-context learning**   We use fine-tuning to adapt our pretrained models to the compositional tasks. Due to its computational cost and task-specificity, fine-tuning is less useful in practice than in-context learning as model size grows (Brown et al., 2020). Because in-context learning only becomes reliable at scales far larger than we are able to train, we did not explore the effect of depth on compositional generalization accuracy in in-context learning (Si et al., 2023); we point this out as an avenue for future research.

## 6 CONCLUSION

Compositional generalization is essential for interpreting novel sentences. What aspects of the transformer LM architecture contribute to an inductive bias favoring compositional generalization? In a controlled experiment that teases apart depth from total number of parameters, we find that deeper transformers show better compositional generalization (and better language modeling performance) independent of their total number of parameters, though in most cases the usefulness of adding layers decreases rapidly as models get deeper. Most of the benefits of depth come from having just a few layers, allowing comparatively shallow models to achieve levels of generalization accuracy on compositional tasks comparable to much deeper models, or to reach language modeling perplexity within a few percentage points of the best-in-class model. We also show the benefits of depth for compositional generalization are not merely a consequence of the fact that deeper models learn the in-distribution data or pretraining corpus better; rather, depth affects generalization over and above these other factors. Our results are robust across nearly an order of magnitude in model size (41M, 134M and 374M parameters).

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

## A  Full table of results

Table 3 displays pretraining and compositional generalization accuracy on all model sizes and tasks.

| $n_{\text{layers}}$ | 41M | | | | | | | 134M | | | | | | | 374M | | | | | | | |
|---|---|---|---|---|---|---|---|---|---|---|---|---|---|---|---|---|---|---|---|---|---|---|
| | 1 | 2 | 3 | 4 | 5 | 6 | 7 | 1 | 2 | 6 | 8 | 12 | 16 | 21 | 1 | 2 | 4 | 6 | 8 | 12 | 16 | 24 |
| C4 val. PPL (↓) | 43.9 | 30.0 | 28.0 | 27.5 | **27.4** | 27.7 | 28.2 | 31.7 | 21.1 | 17.4 | 17.0 | **16.9** | 16.9 | 17.0 | 26.5 | 17.3 | 14.7 | 14.0 | 13.7 | 13.4 | 13.2 | **13.1** |
| COGS (↑) | 12.4 | 58.2 | 63.1 | 68.5 | 63.4 | 68.4 | **72.3** | 19.4 | 65.5 | 74.3 | 72.9 | 73.0 | 75.0 | **75.1** | 21.5 | 66.2 | 72.4 | 75.1 | 75.2 | 76.3 | 76.3 | **78.0** |
| COGS-vf (↑) | 25.7 | 78.3 | 80.8 | 82.5 | 82.5 | 82.6 | **83.0** | 26.3 | 83.0 | 83.2 | 83.7 | 84.7 | 83.8 | **84.8** | 36.8 | 82.2 | 71.9 | 83.1 | 82.6 | 84.3 | **85.1** | 83.1 |
| GeoQuery Standard (↑) | 68.2 | 76.4 | **79.6** | 78.6 | 76.8 | 77.5 | 77.1 | 72.5 | 81.4 | 80.0 | 73.6 | **82.9** | 81.1 | 80.0 | 72.9 | 80.7 | 80.0 | 78.2 | 80.7 | 80.0 | 81.1 | **83.2** |
| English Passivization (↑) | 0.00 | 9.88 | 26.2 | 28.0 | **89.9** | 74.1 | 78.3 | 0.00 | 29.9 | 49.4 | 91.9 | 87.1 | **93.2** | 88.1 | 0.00 | 13.6 | 89.8 | 18.8 | 84.3 | 81.0 | 87.2 | **89.6** |

Table 3: Validation perplexity on C4 after pretraining & generalization accuracy (%) on compositional datasets after 10 k steps of fine-tuning. Bold values indicate best-in-size-class performance.

## B  Annotated transformer layer

Figure 8 shows the schematic for a single transformer layer. The layers input enters on the left and passes through the various model components (grey boxes), being combined with the residual connections before exiting right to subsequent layers. Blue boxes show the dimensionality of the vectors after transformation; we are primarily concerned with the size of the embedding vectors $d_{\text{model}}$ and the internal dimension of the feed-forward block $d_{\text{ff}}$. The size of the vectors internal to the attention mechanism, $d_{\text{attn}}$, is not shown here but is usually set to be equal with $d_{\text{model}}$; we follow this convention here. Non-learned operations like addition, layer normalization, and the feed-forward network's nonlinearity are shown in grey circles.

## C  Feed-forward rank analysis

To investigate the role that the feed-forward block plays in the poor performance of models with extreme $d_{\text{model}}/d_{\text{ff}}$ ratios, we conduct rank analysis on the two transformations $\mathbb{R}^{d_{\text{model}}} \to \mathbb{R}^{d_{\text{ff}}}$ and $\mathbb{R}^{d_{\text{ff}}} \to \mathbb{R}^{d_{\text{model}}}$ which make up the feed-forward block. Our first approach is to conduct singular-value decomposition on each transform. For a given affine transform $T$, we compute the ordered singular values $\{\sigma_1, \sigma_2, \ldots, \sigma_k\}$ where $k = \min(d_{\text{model}}, d_{\text{ff}})$ is the rank of $T$ and $\sigma_i \geq \sigma_{i+1}$. We then normalize each singular value by dividing by the $\ell_1$ norm of $\{\sigma_1, \sigma_2, \ldots, \sigma_k\}$ to calculate how much of the $T$'s image is accounted for by the best $i$-rank approximation of $T$ for $i \leq k$. We note that

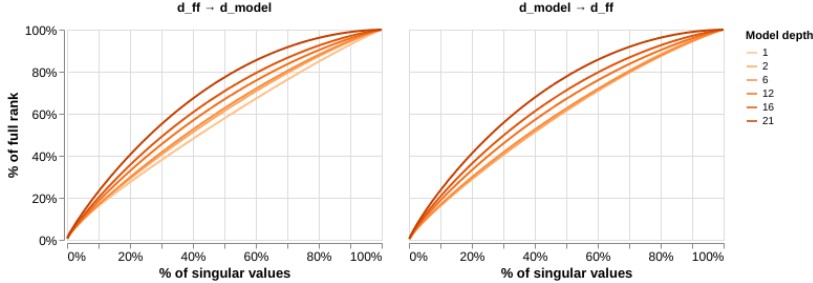

Figure 8: Diagram of a single transformer layer, annotated with the dimensions (blue) of each vector. Information is passed from left to right, through each component (grey box), and added back to the residual embeddings before normalization.

as models get deeper (and consequently, $d_{\text{ff}}$ and gets smaller and the feed-forward ratio $d_{\text{model}}/d_{\text{ff}}$ gets larger), the two transforms in the feed-forward block become increasingly skewed away from making full use of their available ranks (Figure 9).

Figure 9: As models get deeper and $d_{\text{ff}}$ gets smaller, the input **(left)** and output **(right)** projections in the feed-forward block become increasingly close to rank-deficient transforms. A graph of $y = x$ here would indicate that models spread their rank equally across all singular values.

We also measure the *effective rank* of each transform, defined by Roy & Vetterli (2007) a real-valued extension of rank to measure the effective dimensionality of transforms which are close to being rank-deficient:

$$\text{erank}(T) = \exp\left(-\sum \frac{\sigma_i}{\|\sigma\|_1} \log\left(\frac{\sigma_i}{\|\sigma\|_1}\right)\right).$$

We similarly note that the effective rank of the feed-forward transforms decreases as models get deeper and $d_{\text{ff}}$ gets smaller relative to fixed $d_{\text{model}}$, suggesting that our deeper models are increasingly rank-deficient (Figure 10).

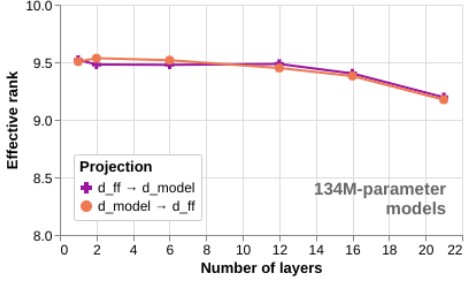

Figure 10: The effective rank of each feed-forward projection, averaged over all layers, decreases as models get deeper and $d_{\text{ff}}$ gets smaller in proportion to $d_{\text{model}}$. A small amount of vertical jitter has been added to help distinguish lines.

