# 1 Compute Efficiency

Reviewers were interested in exploring the compute profiles of models as we vary their depth. We show in Figure 1 that depth strongly influences latency: 32-layer 374M-parameter models are roughly twice as slow as 4-layer 374M-parameter models when running on the same platform during training.

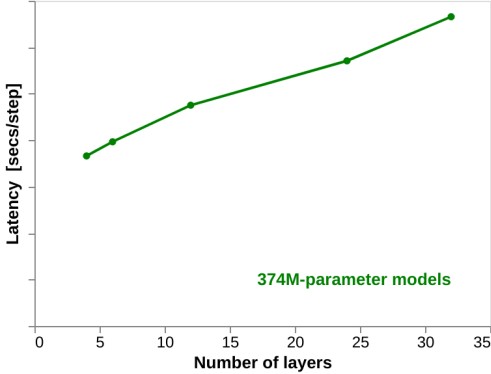

Figure 1: Deeper models are slower than shallower models when controlling for total parameter count. We report relative latency [seconds per step] for 374M-parameter models, showing a strongly-linear relationship between depth and latency.

We note two implications of this finding which address reviewers' questions, in the context of our main result that the marginal utility of depth decreases rapidly as models get deeper:

1. Since models with relatively few layers can attain withing 1% of the performance (i.e., loss for language modeling or generalization accuracy for compositional generalization tasks) of the best, deeper models when trained on equal volumes of data, shallow models can be used at inference-time which have nearly-equal performance as the best models at substantially lower compute costs.

2. Since shallow models have lower latency during training as well as during inference, one can either (a) train a shallower model in much less time than a deeper model on equal volumes of data, or (b) train a shallower model on much more data than a deeper model can be trained on given a fixed compute budget (e.g., access to a particular GPU platform for a given amount of time). Depending on the depths, sizes, volumes of data, and training times involved, this can result in shallower models attaining better performance than deeper models can.

# 2 Replicating results with multiple runs

Reviewers also noted that our results would be stronger if we completed more than a single trial per condition. We are in the process of replicating each result in the

paper with 5 trials per condition. We have finished replicating all pretraining runs, and in Figure 2 show that our results on language modeling hold after replication.

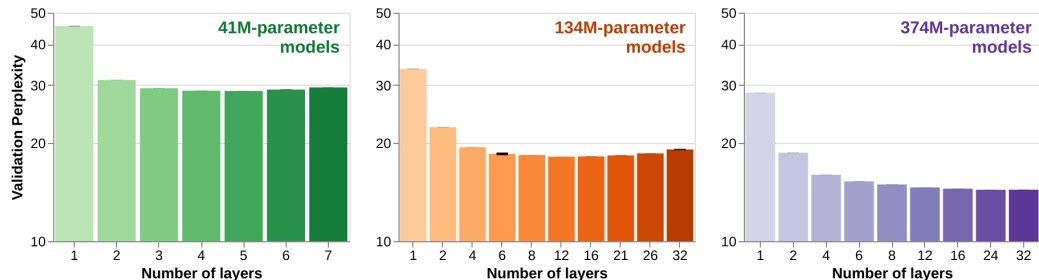

Figure 2: Mean validation perplexity of models by depth after pretraining (132B tokens). Error bars are shown for all sizes & depths, though they are only easily visible on the 6- and 32-layer 134M-parameter models.