# OpenReview forum: "The Impact of Depth and Width on Transformer Language Model Generalization"
_ICLR.cc/2024/Conference — ICLR 2024 Conference Withdrawn Submission_

### Official Review · Reviewer_mDPp · 2023-10-29

**Soundness:** 3 good
**Presentation:** 3 good
**Contribution:** 2 fair
**Rating:** 5
**Confidence:** 3

**Summary:**

This paper studies the effect that LLMs’ depth has on its performance on compositional genalization and language modeling tasks. To disentangle the effect of depth on performance from other factors, the authors fixed the total number of parameters of LLM by reducing the size of the transformer’s feed-forward dimension (d_model) while increasing the layers of transformer. Experiments here showed that deeper LLMs result in better language modeling and compositional generalization up until when the LLM becomes too narrow when d_ff<d_model. The authors also conducted more experiments to show that the better compositional generalization by deeper LLMs is not simply due to better language modeling performance by using pretrained deeper LLMs that have similar perplexity than the shallower counterpart. Experiments are conducted on 3 model size classes, pretrained on C4 corpus and 4 compositional generalization tasks (COGS, COGS-vf, GeoQuery, English passivization).

**Strengths:**

+The paper’s empirical findings contributes to the body of work that seek to better understand how to train LLMs most efficiently by choosing the best mix of model hyperparameters given a particular computational budget.

+Experiments are designed well to disentangle possible confounders (language modeling performance etc).

+The paper is generally well-written and easy to follow.

**Weaknesses:**

-The paper’s core contributions centers around mostly confirming existing findings (e.g. Mueller et al. (2022) and Tay et al. (2021)) with empirical results that a bigger depth improves expressiveness of neural network or LLMs, limiting the impact of the work. Making it more obvious what is different from these prior work will help readers better appreciate the paper’s contributions (e.g. in-depth analyses about why this occurs beyond empirical results on performance).

-The experiments focus only on compositional generalization and language modeling tasks while there is a plerotha of other tasks that can be used to evaluate LLMs’ generalization capabilities.

**Questions:**

Can compositional generalization and language modeling tasks along stand to evaluate the generalization of the LLMs (or mostly only compositional generalization)? It would be helpful to discuss the different types of generalization if the paper is claiming generalization as a whole beyond compositional generalization.


==Post-Rebuttal==
I appreciate the authors' response and decided to keep my score.

---

> ### Author Response · Authors · 2023-11-17
>
> We thank the reviewer for their constructive feedback.
>
> **W. 1: Paper mostly confirms existing findings.**
>
> In our view, prior empirical findings are suggestive but incomplete as the experiments were not controlled: depth was confounded with parameter count. The primary thing which distinguishes our work here from the findings of, e.g., Mueller et al (2022) or Tay et al. (2021) is our goal of controlling for parameter count; while these previous works note that depth can be helpful, this conclusion is fundamentally confounded by the fact that deeper models in their setups are also larger models. Our core contribution is to disentangle that confound and show what the impact of depth is, independent from size. Because, as we show, the usefulness of depth saturates extremely quickly, our results suggest that adding depth does not confer the strong advantage that conventional understanding might expect; rather, adding depth does very little for all but the shallowest models, and in fact, past a certain point increasing model depth can actually be harmful. To our knowledge, these takeaways are novel. Finally, we also show that this effect is present in models’ compositional generalization abilities, and demonstrate that this effect scales with model size.
>
> We have also since conducted an analysis on the compute efficiency of our models by depth. We find that compute efficiency–as measured by latency–is very linear in depth, indicating that models pay a constant price for each additional layer added. Combining this with our main results showing the diminishing returns on depth in terms of pure performance, our results additionally suggest that practitioners bound by resource constraints should prefer to make models much wider and shallower than convention would otherwise suggest. A revised version of the manuscript will include the specifics, and we will include a supplementary figure in our reviewer responses in the next couple of days.
>
> **W. 2: Adding other tasks.**
>
> We agree, and are in the process of evaluating our models on the BigBench-Lite suite of tasks to get a broader picture of how depth impacts model performance. A revised version of the manuscript will include a breakdown of performance-by-depth on these tasks, similar to our section on compositional generalization.
>
> **Question: Generalization beyond compositionality**
>
> In this particular study, we focused on compositional generalization; as we say in the second paragraph of the introduction, “we test the hypothesis that increasing a transformer’s depth—the number of layers it
> has—improves its out-of-distribution performance on tasks that require compositional generalization”. We cite theoretical and empirical studies that motivate this hypothesis. We do not have specific hypotheses as to the effect of depth on other types of generalization, though we agree that it would be interesting to study any such effects empirically in future work. We do agree with you that our scope can be made clearer in the paper; in fact, to make it clearer that we’re only making claims about compositional generalization, as opposed to other types of generalization, we have decided to add the word “compositional” to the title of the paper. We will also go through the manuscript and ensure that the scope of our claims is appropriate.

---

### Official Review · Reviewer_1ZSK · 2023-10-30

**Soundness:** 3 good
**Presentation:** 3 good
**Contribution:** 2 fair
**Rating:** 5
**Confidence:** 2

**Summary:**

The paper provides a controlled study disentangling model depth from the width and total parameters. The results support the view that depth improves generalization in transformers, with diminishing returns past a shallow threshold. The paper makes a solid contribution to understanding model architecture choices for generalization. Overall, the paper makes a valuable contribution by investigating the impact of depth on the generalization ability of Transformer language models. However, addressing the following weaknesses would enhance the comprehensiveness and applicability of the research.

**Strengths:**

By investigating the effect of depth on the model's generalization ability, the paper provides a valuable reference for improving and optimizing the design of language models.

**Weaknesses:**

1. Since the paper mainly verifies the effect of the Transformer’s “depth” on combinatorial generalization, the "depth and width" in the title of the paper is misleading.
2. While the paper primarily investigates the effect of depth, the impact of width on generalization is not extensively explored. It would be beneficial to analyze the trade-offs between depth and width and how they interact in terms of model performance and generalization.
3. The paper does not thoroughly discuss the computational implications of increasing depth or width in Transformer models. Considering the computational cost associated with deeper models, it would be useful to analyze the trade-off between improved generalization and increased computational requirements.

**Questions:**

1. Please double-check the reference format and standardize it.
2. In the paper, you focus on the impact of depth on Transformer language model generalization, but the analysis of width is relatively limited. Can you provide further insights into the trade-offs between depth and width? How do these two factors interact in terms of model performance and generalization? It would be helpful to explore the joint effects of depth and width and their relative importance in achieving better generalization.
3. You only conduct a single run for each experimental condition. Adding multiple runs would strengthen conclusions by quantifying uncertainty and ruling out run-specific fluctuations. Is it feasible to do multiple runs, even if a subset of conditions?
4. Is there an optimal depth where returns diminish for all model sizes and domains? Or does optimal depth keep increasing with the model scale?

---

> ### Author Response · Authors · 2023-11-17
>
> We thank the reviewer for their constructive feedback!
>
> **W. 1 The title of the paper should focus on depth, not width.**
>
> Thank you for this comment! On reflection, we tend to agree, and have decided to change the title as you propose.
>
> **W. 2: The effect of width is not extensively explored.**
>
> We agree, and this is intentional: the goal of the paper is to test the hypothesis that deeper models have stronger compositional generalization abilities. To the extent that we do vary width, we do so only to enable fair comparisons across models of different depths (by keeping the total number of parameters matched). While there may be interesting effects of width (and other hyperparameters) on generalization, we do not have particular hypotheses as to what those would be, and as such such experiments are outside the scope of the paper. We hope that by changing the title we have made the scope of the paper clearer.
>
> **W. 3: Compute performance analysis.**
>
> We fully agree that studying the compute impact of depth in addition to the performance impact is important. Since submitting the original manuscript, we have conducted an analysis of the compute cost of increasing depth as measured by latency; we plan to include this analysis in a future version of the manuscript very soon. We find that latency is roughly linear in depth, indicating that models incur a consistent compute-efficiency penalty as depth increases. This, combined with our main results which show that the performance improvement depth affords diminishes rapidly and quite early suggests that compute-constrained teams should likely prefer shallower models which are just deep enough to capture a substantial portion of the benefits depth affords, for two reasons. (1) In training, shallower models will either train in a shorter amount of time (for a fixed data budget) or can train on more data (for a fixed time budget) than deeper models of the same size; (2) In inference, shallower models have lower per-use latencies than deeper models of the same size do.
>
> **Q. 1: Standardize citation format.**
>
> Thanks for pointing this out, we have gone through the bibliography and standardized our references.
>
> **Q. 2: The tradeoff between depth and width.**
>
> As we mention in our response to Weakness 2, the main focus of our paper is on the effect of depth on generalization. To study it while keeping the comparison fair, however, we do need to trade depth off against width: for all models within a size class, we have to vary depth and width inversely with one another, which allows us to quantify how different depth-width combinations (for a fixed number of parameters) fare on our studied tasks. One particularly interesting finding we report concerning this tradeoff is the U-shaped relationship between depth and perplexity: when models become very deep their performance begins to degrade; as we mention in the paper, we hypothesize that this is due to the fact that they become too narrow.
>
> **Q. 3: Performing additional runs.**
>
> Thanks for pointing this out - we fully agree, and have been working to replicate our analyses on multiple runs. We have updated our paper to replicate each result five times (that is, we now have 5 pretraining runs per condition for the language modeling results, and we have 5 fine-tuning runs based on those 5 pretraining checkpoints for the compositional generalization results). We have also added confidence intervals to both of these sets of results to quantify measurement uncertainty. In all cases, we find that our initial results hold up after repeated trials, strengthening our confidence in the conclusions.
>
> **Q. 4: Is there an optimal depth for all sizes & domains?**
>
> Thanks for this very insightful question. Our analysis in Section 3.1, particularly the data shown in Fig. 3(b), suggests that the optimal performance–as measured by absolute score for a fixed data budget–consistently appears to be when the ratio of feed-forward dimension to embedding dimension is between 0.5 and 1.0. In our setup, this would mean that the optimal depth for a model should in fact increase with size, assuming that the embedding dimension is also scaled with parameter count, although this point may not line up with the depth of conventional existing models. For instance, among the model families we examine, the optimal 41M and 374M models are deeper than their conventional reference models. However, taking into account the reviewer’s earlier point about compute cost, the “compute-optimal” depth may actually be much shallower than convention suggests.

---

### Official Review · Reviewer_yaYa · 2023-11-01

**Soundness:** 3 good
**Presentation:** 3 good
**Contribution:** 2 fair
**Rating:** 3
**Confidence:** 4

**Summary:**

The paper empirically studied the impact of increasing depth and width on the model's out-of-distribution generalization performance.

**Strengths:**

1. The paper provides some interesting experiment results which might be useful for future research.

**Weaknesses:**

1. The result is a bit too straightforward with only experiment results. More theoretical analysis on the difference between increasing depth and width on out-of-distribution generalization is required for a paper on venues such as ICLR.
2. Why do the authors choose to focus on decoder-only models? What can be the difference between encoder-decoder models and decoder-only models on the impact from different depths and widths?

------- post rebuttal ------
I have read the rebuttal.
Although empirical study is also important for machine learning research, the paper lacks some insightful new information for the community.
The rating should be between 3 or 5 but there isn't an option for 4. I'll keep my rating.

**Questions:**

Please see the weakness part.

---

> ### Author Response · Authors · 2023-11-17
>
> We thank the reviewer for their helpful comments.
>
> **W1. The paper focuses on empirical experiments**
>
> We agree that this paper is an empirical experimental study, though it is motivated by a theory (Merrill, Sabharwal & Smith, TACL, 2022), but we disagree that this is a reason to reject the paper; in our experience, ICLR regularly publishes papers that are primarily experimental, and in fact such papers may be more common than theory-heavy papers. In general, we believe that there is a place for both theory papers and empirical papers, and especially for empirical papers such as ours that test predictions made by a theory.
>
> **W2. The focus on decoder-only transformers**
>
> We agree with the reviewers that in future work it would be worthwhile to investigate additional architectures, including encoder-decoder transformers or even LSTMs, but given the vast amount of computational resources required to pretrain language models, we had to limit our scope to a single architecture. We chose to focus on decoder-only transformers as at the time of writing they were the state-of-the-art and the de-facto standard for both proprietary (PaLM, GPT-3,4) and open-source (GPT-2, LlaMA) large language models.

---

### Official Review · Reviewer_ZnHf · 2023-11-03

**Soundness:** 4 excellent
**Presentation:** 4 excellent
**Contribution:** 2 fair
**Rating:** 6
**Confidence:** 3

**Summary:**

This paper  studies how performance for "compositional generalization" in Transformers varies as a function of depth. Its main twist is to pay careful attention in keeping the number of parameters constant. Hence, when augmenting depth, it is reducing width accordingly. This is done for 3 different number of parameters.
Inspecting the results, my take-aways are the following: performance _does_ systematically get better with deeper models as long as they don’t become so narrow so as to have a width that requires reducing input dimensionality. This said, depths=3-6 look largely enough for all practical purposes, and the main way to get better performance is just to increase the number of parameters, which matches usual knowledge.

**Strengths:**

The paper asks a clear question: how does performance vary as a function of depth vs width for a given and fixed number of parameters for transformer based architectures on LM.
The paper provides a very clear and rigorous treatment of this question, also providing relevant literature and areas of further investigations.
I particularly like one of the final questions that is asked about "alternative approaches to controlling for total size". Universal transformers are quite an extreme way to go, with all layers sharing the same weights. Maybe you could find some alternative way, for instance by repeating blocks of layers instead of just one. Likewise, I wonder about hypernetworks. They could be used to fill out huge networks, but then constraining the number of parameters.

All in all, I think the paper may be interesting to some persons, at least as a reference on that precise question it is asking.

I think the paper is just as good as it gets to answer this question for any person that could be interested in the topic. For this, my pick is it should be accepted.

**Weaknesses:**

- Applicability of the study is arguable a bit weak and I would say that it mostly would serve as a reference for what is usually considered common knowledge without any rigorous treatment: "for a given parameter budget, pick depth over width".
- It remains extremely clear from this paper that beyond very small depth (as soon as we get 3~ layers), performance doesn’t really go up with depth alone: the way to go is just to add more parameters.
- As a practitioner, I would be interested by the following question: what about if my budget is not really in terms of number of paremeters, but rather in compute power or memory? Do you see the same thing happening that one should pick depth?
- p8: "when studying the the impact"

**Questions:**

I am not sure about the questions I should ask, since the paper really looks pretty clear to me. I guess it’s more about what’s next. Personally I didn’t find the 3rd and 4th limitations very illuminating, but liked the 2nd.

---

> ### Author Response · Authors · 2023-11-17
>
> We thank the reviewer for their constructive feedback.
>
> **W1: The study would mostly serve as a reference for what is usually considered common knowledge**
>
> The reviewer’s summary of our results as a recommendation to, for a given parameter budget, pick depth over width, has some truth in it, but it does not fully convey the richness of results we report. First, we find that the marginal utility of added depth reduces significantly as models get deeper, such that a more accurate summary of our recommendation would need to be amended by the statement _“for a given parameter budget, picking a very deep model will not help much over picking a ‘just-deep-enough’ model”_. But we also find that when the corresponding feed-forward block becomes too small, it can actually be harmful to increase depth. Finally, motivated by prior theoretical claims, we show empirically that depth dramatically affects models’ compositional generalization abilities, and demonstrate that this effect scales with model size; we are aware of very little work that has investigated the sensitivity of compositional generalization to depth. Overall, then, we believe that our results significantly expand upon what anyone might take to be “common knowledge” in the field. Even if that were not the case, however, in our view there is significant value in publishing controlled scientific experiments that confirm a commonly but informally belief.
>
> **W2: It remains extremely clear from this paper that beyond very small depth (as soon as we get 3~ layers), performance doesn’t really go up with depth alone**
>
> The diminishing return to increasing depth is indeed one of the interesting findings of our paper (though the specific number of layers where we begin to observe diminishing returns varies by task and is usually greater than 3). But we are not sure why the reviewer mentions this finding as a weakness of the paper - this is an empirical scientific discovery, one we didn’t expect to find! This discovery also has applied implications: standard pretrained transformers may be deeper, and therefore more computationally expensive at inference time, than they need to be to generalize well (see also the next point).
>
> **W3: I would be interested by the following question: what about if my budget is not really in terms of number of paremeters, but rather in compute power or memory?**
>
> Great question, thank you for pointing this out. We had already begun investigating this after the submission deadline and we are able to provide an answer. A revised version of the manuscript will include an analysis of the compute performance by model depth. For the accelerators we trained on, we find a strongly-linear relationship between depth and latency when controlling for parameter count. Because latency increases linearly with depth while performance saturates quite early, compute-constrained practitioners can benefit from choosing architectures which are “just deep enough” for two reasons: (1) since shallower models are computationally cheaper, they can be trained in less time for a fixed data budget, or trained on more data for a fixed time budget; (2) for the same reason, shallower models incur less per-inference compute cost. This reduces point-of-use latency.
>
> Taken together, these points actually point against what commonly-understood knowledge might otherwise suggest: _picking depth over width is not necessarily the best choice_. Rather, if one can ensure that a model is deep enough to capture most or all of the value of added depth (which happens quite early on), practitioners concerned with things like compute or latency tradeoffs should actually then choose to make models as wide as possible.
>
> **W4: p8: "when studying the the impact"**
>
> Thank you for catching the typo. It has been fixed.

---

### Author Response · Authors · 2023-11-23
**General Comment**

We thank the reviewers for their comments and constructive feedback, which has helped us improve the paper. Reviewers noted that our paper addresses a useful gap in existing literature and provides a clear point of reference for the (diminishing) utility of depth for language modeling and compositional generalization. Our results partly support theoretical predictions on the expressive capacity of transformer models, and can help inform the design of compute-efficient language models.

Reviewers also raised a number of concerns which we have since addressed:
- **Number of runs:** Our original paper reported only a single trial per condition. We have since completed a complete replication of all pretraining runs in the paper (5 trials per condition) and are in the process of completing a replication of all fine-tuning runs. We note that all replicated results have been entirely consistent with our original findings. We have included in supplementary materials a version of the language modeling results showing error bars for the full results with multiple pretraining runs.
- **The compute requirements of deep vs. shallow models:** The reviewers correctly pointed out an important consequence of our findings: because compositional generalization accuracy plateaus after a relatively modest number of layers, and because compute costs may scale linearly with depth, it may be more compute-efficient for transformers to be shallower than is currently standard. We have included in supplementary materials an analysis of the compute profiles of models by depth, as measured by latency. We note a strongly linear trend which, combined with our existing results on the diminishing utility of depth, suggests that indeed compute-constrained groups should prefer much shallower models than current convention would suggest to attain compute-optimal performance.
- **The focus on compositional generalization:** Our study tests a theoretical hypothesis that is specific  to compositional generalization; we have made this focus clearer in the manuscript (for example by changing the title).  At the same time, we agree that fine-tuning the models we have pretrained on other tasks could potentially reveal unexpected empirical patterns; we will include in an updated manuscript additional experiments on a variety of other LLM evaluation tasks from the BIG-bench suite.